# Assessment of Vitamin D Metabolism Disorders in Hemodialysis Patients

**DOI:** 10.3390/nu17050774

**Published:** 2025-02-22

**Authors:** Maksymilian Hryciuk, Zbigniew Heleniak, Sylwia Małgorzewicz, Konrad Kowalski, Jędrzej Antosiewicz, Anna Koelmer, Michał Żmijewski, Alicja Dębska-Ślizień

**Affiliations:** 1Department of Nephrology, Medical University of Gdańsk, Dębinki 7 Street, 80-211 Gdańsk, Poland; maxhry1308@gumed.edu.pl (M.H.); sylwia.malgorzewicz@gumed.edu.pl (S.M.); alicja.debska-slizien@gumed.edu.pl (A.D.-Ś.); 2Department of Clinical Nutrition, Medical University of Gdańsk, Dębinki 7 Street, 80-211 Gdańsk, Poland; 3Department of Bioenergetics and Exercise Physiology, Medical University of Gdańsk, Dębinki 1 Street, 80-211 Gdańsk, Poland; konrad.kowalski@masdiag.pl (K.K.); jedrzej.antosiewicz@gumed.edu.pl (J.A.); 4Masdiag Laboratory, S. Żeromskiego 33 Street, 01-882 Warsaw, Poland; 5Centre of Biostatistics and Bioinformatics Analysis, Medical University of Gdańsk, 1a Dębinki, 80-211 Gdańsk, Poland; anna.koelmer@gumed.edu.pl; 6Department of Histology, Medical University of Gdańsk, Dębinki 1 Street, 80-211 Gdańsk, Poland; michal.zmijewski@gumed.edu.pl

**Keywords:** vitamin D metabolite ratio, 3-epi-25(OH)D3, 24,25(OH)2D3, 25(OH)D3, 25(OH)D2, 1,25(OH)2D3, vitamin D, chronic kidney disease, hemodialysis, alphacalcidiol

## Abstract

Background: Patients with end-stage chronic diseases, especially those undergoing hemodialysis (HD), often experience mineral bone disease (MBD), leading to hypocalcemia, hyperphosphatemia, and elevated parathyroid hormone (PTH). Vitamin D deficiency and metabolism disorders are also common, resulting from impaired conversion of 25(OH)D3 to its active form, 1,25(OH)2D3, and reduced inactivation to 24,25(OH)2D3. This study aimed to assess the levels of 25(OH)D2, 25(OH)D3, 24,25(OH)2D3, 3-epi-25(OH)D3, and the vitamin D metabolism ratio (VMR) in patients with maintenance HD. Methods: A cross-sectional study was conducted on 66 HD patients (22–90 years, average 61.3 ± 16.4), with a control group of 206 adults without chronic kidney disease (CKD), both without cholecalciferol supplementation. Results: the HD patients had significantly lower 25(OH)D3 levels (15 ng/mL vs. 22 ng/mL) and higher deficiency rates (69% vs. 39%) compared to the controls. However, both groups showed similarly low levels of optimal vitamin D3. The HD patients had lower 24,25(OH)D3 levels (0.1 vs. 2.1 ng/mL) and a lower VMR (0.9% vs. 9%). 3-epi-25(OH)D3 levels and its ratio to 25(OH)D3 were significantly lower in the HD group. Alphacalcidol supplementation raised 1,25(OH)2D3 levels (30.4 vs. 16.2 pg/mL) without affecting other vitamin D metabolites. The HD patients had higher levels of 25(OH)D2 compared to the controls (0.61 vs. 0.31 ng/mL). Conclusions: Vitamin D3 reserves are lower, and both functional deficiency and impaired catabolism of vitamin D3 are present in HD patients compared to the general population. The VMR index is the most sensitive parameter for vitamin D3 deficiency assessment, highlighting the importance of measuring 24,25(OH)D3. Alphacalcidol supplementation increases 1,25(OH)2D3 levels without affecting other vitamin D metabolites. 25(OH)D2 is the only metabolite that was higher in HD patients than the controls.

## 1. Introduction

Vitamin D (ergocalciferol and cholecalciferol) describes a group of fat-soluble secosteroids that maintain bone health by regulating calcium and phosphate metabolism. The active form of vitamin D, calcitriol (1,25(OH)2D3), increases renal phosphate reabsorption, as well as intestinal phosphate and calcium absorption [1]. It is estimated that optimal levels of vitamin D (in optimal level) enhance calcium levels by 30–40%, and phosphate absorption by 80% [2].

Cholecalciferol (vitamin D3) is synthesized in skin cells from 7-dehydrocholesterol as a result of ultraviolet B radiation (UVB). It is then converted by hepatic 25-alpha-hydroxylase into calcifediol (25(OH)D3). Ergocalciferol (vitamin D2) does not undergo photoisomerization in skin cells. Its main sources include plant-based products, mushrooms, and dietary supplements [3].

Calcifediol (25(OH)D) obtained from both vitamin D2 and D3 is hydroxylated by 1-alpha-hydroxylase to 1,25(OH)2D3 in the proximal convoluted tubule [4]. Through a feedback inhibition mechanism, 1,25(OH)2D3 limits its own production by inhibiting CYP27B1 and upregulating the expression of CYP24A1. The latter encodes 24-hydroxylase, responsible for inactivating calcitriol to 24,25-dihydroxycholecalciferol [5]. In patients with chronic kidney disease (CKD), the level of 1,25(OH)2D3 is significantly reduced in proportion to the stage of kidney damage. This is mainly caused by the gradual loss of nephrons and an increased level of fibroblast growth factor-23 (FGF-23), which decreases the activity of 1-alpha-hydroxylase [6,7]. As a result of the worsening calcitriol deficit, secondary hyperparathyroidism (SHPT) develops, especially in dialyzed patients. CKD affects over 10% of the global population, accounting for more than 800 million individuals [8], of whom 4 million receive renal replacement therapy, with 70% undergoing hemodialysis (HD) [9,10].

The assessment of vitamin D status is essential for the prevention and treatment of SHPT. 25(OH)D3 is primarily used to evaluate the body’s vitamin D3 reserves and its adequate intake, serving as a relatively inflexible marker that does not adequately reflect functional disturbances in calcium–phosphate metabolism, unlike 1,25(OH)2D3 [11]. However, due to its short half-life and very low serum levels, KDIGO still recommends using 25(OH)D3 to assess vitamin D3 levels in patients with CKD, as in the general population [12,13]. In light of this information, it would be beneficial to develop a compound that combines the advantages of the aforementioned metabolites while avoiding their defects.

24,25(OH)2D3 is a promising biomarker for assessing vitamin D metabolism activity, particularly in patients with CKD. The vitamin D metabolite ratio (VMR), defined as the proportion of 24,25(OH)2D3 to 25(OH)D3, provides a more accurate measure of vitamin D adequacy than 25(OH)D3 alone, as it is less influenced by confounding factors like variability in vitamin D binding protein (VDBP) levels [14]. CYP24A1 (24-hydroxylase), the key enzyme responsible for catabolizing 25(OH)D3 and 1,25(OH)2D, plays a central role maintaining plasma vitamin D homeostasis through a negative feedback mechanism. Its transcription is induced by high levels of calcitriol and FGF-23, but it is suppressed by parathyroid hormone (PTH) [15]. In CKD, reduced CYP24A1 activity is driven by low calcitriol, high PTH, decreased renal mass, and an impaired delivery of 25(OH)D3 to tubular cells. This leads to lower 24,25(OH)2D3 levels despite 25(OH)D3 concentrations being comparable to the general population. Consequently, dialysis patients should exhibit very low VMR values [15,16,17]. Notably, higher VMR levels are correlated with reduced risks of fractures, slower CKD progression, and lower overall mortality, correlations not observed with 25(OH)D3 levels. This further underscores its clinical relevance [14].

The metabolite 3-epi-25(OH)D3, also referred to as the 3-epimeric form of 25-hydroxyvitamin D3, is a structural variant of 25(OH)D3, differing only in its stereochemistry at the C3 position. This epimer arises through a process of epimerization and exhibits unique biological characteristics. Compared to 25(OH)D3, 3-epi-25(OH)D3 has a diminished capacity to increase calcium levels in the blood and has less potent effects on genes sensitive to the vitamin D receptor. Nevertheless, it retains comparable efficacy to 1,25(OH)2D3 in suppressing PTH gene transcription, suggesting a potential protective role against hypercalcemia in cases of hypervitaminosis D3. Current data suggest that high vitamin D levels do not significantly affect the percentage of 3-epi-25-OH-D3, despite high absolute concentrations [18]. The studies suggest that the ratio of 3-epi-25(OH)D3 to 25(OH)D3 may provide valuable insights for a more precise assessment of vitamin D status [19].

It seems reasonable to identify a better indicator for the evaluation of vitamin D activity in the CKD population, especially in dialyzed patients, to improve diagnostic approaches and treatment outcomes. The aim of this study was to demonstrate differences in the concentrations of vitamin D metabolites between dialyzed patients and the general population, especially in terms of 24,25(OH)D3 levels and VMR values, as well as epi-25(OH)D3 and epi-25(OH)D3/25(OH)D3, to determine their usefulness in assessing vitamin deficiencies.

## 2. Materials and Methods

We enrolled 66 adults HD patients and 206 healthy participants as the control in this cross-sectional study. All patients provided informed consent to participate in the study. The study received ethical approval No. NKBBN/96/2020, prepared by the Bioethic Committee of Medical University of Gdansk, Poland.

The study group consisted of 66 adult HD patients aged 22 to 90 years, with an average age of 61.3 ± 16.4 years, and 38 (57.6%) were male. Patients were hemodialyzed at the Dialysis Centre of the Department of Nephrology and Transplantology and Internal Diseases of the University Clinical Center in Gdańsk, Poland, between July and December 2021. HD was performed 3 times per week (high-flux dialyzer, 12–14 h per week). The mean Kt/V (parameter to measure dialysis adequacy, evaluating the removal of urea relative to the patient’s total body water) was 1.63 ± 0.29. (reference value Kt/V ≥ 1.2) [20]. Kt/V was calculated according to following formula: K: the clearance rate (mL/min), representing the volume of plasma cleared of urea per minute by the dialyzer; t (time): the duration of a single HD session (typically in hours), V (volume of urea distribution): the estimated total body water volume, usually 55–60% of body weight for adults.

Twenty-five (37.9%) of the patients, due to SHPT, were administered the active form of vitamin D (alphacalcidol) at a mean dose of 0.25 mcg per day for at least 3 months. All HD patients were under the regular care of a renal dietitian. The standard dietary recommendations included a phosphate intake of up to 1000 mg/day, protein intake of approximately 1.0–1.3 g/kg body weight, and potassium intake of 2–2.5 g/day. Additionally, in our center, nutritional status was assessed using the Subjective Global Assessment (SGA) every 6–12 months. None of the patients met the criteria for severe malnutrition, which was an exclusion criterion for the study. In the HD group, the primary causes of CKD were diabetes (38.5%), atherosclerotic nephropathy (16.9%), glomerulonephritis (11.2%), autosomal dominant polycystic kidney disease (7.5%), vasculitis (4.6%), and unknown etiology (21.3%). Additionally, 91.5% of patients suffered from hypertension. The inclusion criteria were as follows: patient consent for participation in the study, a minimum age of 18 years, at least 6 months of HD treatment, regular HD treatment (3 times per week), and a Kt/V value ≥ 1.2. Exclusion criteria included lack of consent, dialysis treatment duration of less than 6 months, having spent more than 5 days in a high-temperature country located in the warm temperate, subtropical, or tropical zone within the last 3 months, diseases: (liver insufficiency, liver disease (hemochromatosis, Wilson’s disease), autoimmune diseases requiring immunosuppressive therapy), severe malnutrition, use of cholecalciferol or other vitamin D forms other than alphacalciol, diseases affecting calcium–phosphate imbalance (sarcoidosis, Fanconi syndrome, Crohn’s disease and ulcerative colitis, multiple myeloma, tumors producing PTHrP, or sarcoidosis), advanced-staged cancer, and liver insufficiency. The details are partly presented in Figure 1 and Table 1.

The control group consisted of 206 randomly enrolled adults without CKD aged from 29 to 86 years old, with an average age of 60.8 ± 13.7 years; 101 (37.1%) of them were males. None of the participants were using any form of vitamin D or any medications that directly affect calcium–phosphorus metabolism, nor were they following any specific diet.

Patients were subjected to a complete history review regarding the usage of vitamin D, and peripheral blood samples were analyzed for the concentrations of vitamin D metabolites, including 25(OH)D2, 25(OH)D3, 24,25 (OH)2D3, 3-epi-25(OH)D3, and 1,25(OH)2D3 in the HD group.

### 2.1. Sample Collection and Measurements of CBC, CRP, Phosphate, and Total Calcium

Blood was taken from the patients’ veins and collected in tubes appropriate for biochemical analysis (e.g., serum or plasma tubes). The blood samples were centrifuged at 1500–2000× *g* for a duration of 10–15 min to separate serum or plasma. The measurements were immediately performed in the hospital laboratory. Total calcium was determined using a colorimetric method, where calcium reacts with a specific reagent (e.g., arsenazo III), resulting in a color change proportional to calcium concentration, which is measured spectrophotometrically. Phosphates were measured using a colorimetric assay, where they react with ammonium molybdate in an acidic environment to form a complex that can be measured spectrophotometrically. Complete blood count (CBC) was performed using an automated hematology analyzer, which applies electrical impedance and flow cytometry methods to determine blood cell counts and morphological parameters. C-reactive protein (CRP) concentrations were measured using an immunoturbidimetric assay, in which CRP reacts with specific antibodies, causing turbidity changes that are measured spectrophotometrically.

### 2.2. Sample Collection and Measurements of Vitamin D Metabolites Levels

Blood (9 mL) was collected once, in the morning. Venous blood samples were collected into S-Monovette^®^ tubes (Sarstedt, Nümbrecht, Germany) containing a coagulation accelerator for serum separation. The serum was processed using standard methods, aliquoted, and stored at −80 °C until analysis.

Sample preparation involved serum protein precipitation and derivatization using DAPTAD (synthesized by Masdiag, Warsaw, Poland). Quantitative analysis was performed via liquid chromatography–tandem mass spectrometry (LC-MS/MS, QTRAP^®^4500, Sciex coupled with an ExionLC HPLC system), with minor modifications based on a previously published method [21]. Raw data were processed with Analyst^®^, and quantification was performed with MultiQuant^® (^Version 3.0.3, Framingham, MA, USA). LC-MS-grade reagents, including acetonitrile (ACN), water, ethyl acetate, methanol, and formic acid (FA), were used.

The analysis measured 25(OH)D3, 24,25(OH)2D3, 3-epi-25(OH)D3, 25(OH)D2, VMR, and 3-epi-25(OH)D3 to 25(OH)D3. Vitamin D metabolite concentrations were corrected for changes in plasma volume [22].

The VMR was calculated by dividing the 24,25(OH)2D3 concentration by the 25(OH)D3 concentration and then multiplying it by 100% [23].

### 2.3. Statistical Analysis

For the statistical analysis, the Shapiro–Wilk test (S-H test) was used to check the normality of the distribution of variables. Since the data did not follow a normal distribution, the median values were reported, and the non-parametric Mann–Whitney U test (M-W test) was employed to assess the statistically significant difference in the vitamin D parameter levels between patients in the control group and the study group. To assess statistically significant differences in the vitamin D metabolite levels in the HD group (between subgroups supplementing alfadiol and not supplementing alphacalcidol), the M-W test was used. To assess the correlation between vitamin D metabolites in the study and control group, Spearman’s rank correlation coefficient was used. A very high correlation strength was defined between 0.7 < rho < 1.0, high correlation between 0.5 < rho < 0.7, medium correlation between 0.3 < rho < 0.5, and low correlation was 0.1 < rho < 0.3. Statistical significance was set at *p* < 0.05.

To assess whether differences in gender distribution between the groups could have influenced metabolite levels, we performed a Mann–Whitney test, treating gender as an independent variable and metabolite levels as dependent variables. The analysis revealed no statistically significant differences in metabolite levels between males and females (*p* > 0.05), indicating that gender differences between the groups did not affect the study outcomes. All calculations were performed using IBM SPSS Statistics (Version 30.0.0, Armonk, NY, USA) and Jamovi (Version 2.6.25, Sydney, NSW, Australia) statistical software for Windows, provided by the Centre of Biostatistics and Bioinformatics Analysis at the Medical University of Gdańsk.

## 3. Results

### 3.1. 25(OH)D3 Levels in Studied Groups

The median level of 25(OH)D3 in the HD group was statistically significantly lower than in the control group (14.6 [9.31–25.27] vs. 22.89 [16.2–29.6], *p* < 0.001) [ng/mL]. This is shown in Figure 2 and Table 2. A suboptimal level (deficiency and insufficiency at <20 and between 20 and 30 ng/mL, respectively) of 25(OH)D3 affected 79.6% of participants in the control group and 81.8% in the HD group. Vitamin D deficiency was observed in 39.8% of the control group and 68.2% of the HD group, while vitamin D insufficiency was found in 35.9% of the control group and 12.1% of the HD group (Table 3).

### 3.2. Vitamin D Metabolites and Vitamin D Metabolites Ratios (24,25(OH)2D3 to 25(OH)D3, epi-25(OH)D3/25(OH)D3) in Studied Groups

The median level of 24,25(OH)2D3 [ng/mL] in the HD group was statistically significantly lower than in the control group (0.1 [0.06–0.31] vs. 2.09 [1.30–3.04], *p* < 0.001) (Figure 3 and Table 2). The median level of the VMR was lower in the HD group in comparison to the control population, as shown in Table 2. The median level of epi-25(OH)D3 [ng/mL] in the HD group was statistically significantly lower than in the control group (0.40 [0.29–0.67] vs. 0.96 [0.52–1.6], *p* < 0.001), as shown in Table 2.

In the patients with HD, the ratio of epi-25(OH)D3/25(OH)D3 (expressed as a percentage) was statistically significantly lower in comparison to the control group (2.77% vs. 4.59%, *p* < 0.001) (Table 2 and Figure 4).

### 3.3. 25(OH)D2 Level in Studied Groups

The median level of 25(OH)D2 [ng/mL] in patients from the HD group was statistically significantly higher than in the control group (0.61 [0.46–0.93] vs. 0.31 [0.21–0.53], *p* < 0.001) (Table 2 and Figure 4). The current literature and guidelines from scientific organizations, such as the Endocrine Society and the National Institutes of Health, do not provide specific reference ranges for 25(OH)D2 [12].

### 3.4. 1,25(OH)2D3 in HD Patients Supplemented with Alphacalcidol

The median level of 1,25(OH)2D3 [pg/mL] in the HD group was 21.4 pg/mL (Table 4). The median level of 1,25(OH)2D3 [pg/mL] in patients supplemented with alphacalcidol was statistically significantly higher compared to those not receiving the drug (30.4 [25.2–56.5] vs. 16.2 [12.8–26.2], *p* < 0.01).

However, there was no statistically significant difference in the median concentrations of the metabolites: 25(OH)D3, 24,25(OH)2D3, 3-epi-25(OH)D3, and the VMR between these patients (Table 4).

Therefore, we compared the metabolites in the HD group to those of the general population, despite some patients using alphacalcidol. This approach was justified as the supplementation did not influence the metabolite concentrations, allowing for a unified analysis across the dialysis cohort.

### 3.5. Correlations Between Studied Parameters

In the present study, no correlation was found between the concentration of vitamin D metabolites and the patient’s gender and age.

There was a statistically significant (*p* = 0.003) positive correlation between the levels of 1,25(OH)2D3 and 25(OH)D3. Spearman’s rank correlation coefficient was 0.383.

There was a statistically significant (*p* < 0.001) positive correlation between the levels of 24,25(OH)2D3 and 25(OH)D3 in the HD group, and Spearman’s rank correlation coefficient for these variables was high, 0.714. Also, in the control group, there was a statistically significant positive correlation (*p* < 0.001) between 24,25(OH)2D3 and 25(OH)D3 levels, with Spearman’s R = 0.885 (Figure 5).

There was no statistically significant (*p* = 0.150) correlation between the levels of 25(OH)D2 and 25(OH)D3 in the HD group, and Spearman’s rank correlation coefficient for these variables was −0.179. Also, in the control group, there was not a statistically significant correlation (*p* = 0.513) between 25(OH)D2 and 25(OH)D3 levels, with Spearman’s R = −0.046.

There was a statistically significant (*p* < 0.001) correlation between the levels of 3-epi-25(OH)D3 and 25(OH)D3 in the HD group, and Spearman’s rank correlation coefficient for these variables was 0.915. Also, in the control group, there was a statistically significant positive correlation (*p* < 0.001) between 3-epi-25(OH)D3 and 25(OH)D3 levels, with R Spearman’s = 0.776 (Figure 6).

There was a statistically significant (*p* < 0.001) positive correlation between the of 24,25(OH)2D3 and 3-epi-25(OH)D3 levels in the HD group, and Spearman’s rank coefficient for these variables was 0.692. Additionally, in the control group, there was statistically significant positive correlation (*p* < 0.001) between 24,25(OH)2D3 and 3-epi-25(OH)D3 levels, with Spearman’s R = 0.780 (Table 5).

## 4. Discussion

As CKD progresses, tubular phosphate excretion is less effective, leading to hyperphosphatemia, which in turn contributes to the increased release of PTH and FGF-23 to maintain normophosphatemia by increasing the phosphaturic effect (this action becomes insufficient starting from stage 4 CKD) [11] and activity of renal CYP24A1, which encodes the 24-hydroxylase, resulting in calcitriol inactivation and acting as a main inhibitor of 1-alpha hydroxylase [5]. Concomitant hyperphosphatemia, reduced renal 1 alpha hydroxylase (CYP27B1) activity, and hypocalcemia (secondary to low calcitriol) are directly responsible for the increased release of PTH, leading to SHPT.

Moreover, SHPT can lead to vascular calcification, contributing to significant cardiovascular mortality in patients with CKD [24], as well as to bone disease, consequently increasing the risk of fractures [25]. Nevertheless, an increased PTH concentration is usually not enough to normalize phosphate and calcium levels; therefore, the most common set of laboratory abnormalities in CKD includes hypocalcemia, hyperphosphatemia and significantly elevated PTH with vitamin D deficiency. The above-mentioned calcium-phosphate metabolism disorders, in the setting of CKD, are referred to as CKD-MBD (chronic kidney disease-mineral bone disorder) [5].

### 4.1. 25(OH)D3

The measurement of 25(OH)D3 is the most reliable indicator of vitamin D status due to its long half-life (3 weeks) [26,27]. While it reflects dietary vitamin D supply, it is not the most active metabolite, nor is its synthesis regulated by feedback mechanisms. This makes 25(OH)D3 less effective in assessing mineral and bone disorders compared to 1,25(OH)2D3 [28]. Despite this, KDIGO recommends its use for evaluating vitamin D status [13]. The National Institute of Health defines vitamin D deficiency (level < 20 ng/mL) and insufficiency as 20 to 30 ng/mL [29]. Based on these thresholds, in our study, suboptimal levels (<30 ng/mL) were common in both the general population and dialysis patients, with deficiency being significantly higher in dialysis patients (68% vs. 40%). The 25(OH)D3 levels in dialysis patients were significantly lower than in the control group.

The HELENA study (over 1000 individuals from nine European countries) reported a median 25(OH)D3 level of 22.4 ng/mL in the general population, similar to our findings [30]. Globally, low vitamin D levels are common, affecting up to 80% of the population, with regional differences (e.g., approx. 60% in Scandinavia and United States.). In Europe, suboptimal levels occur in 80% of the population, with deficiency observed in 40% of the population [31,32,33]. A pooled analysis of studies (2002–2022, 8 million participants) showed levels <30 ng/mL in 70%, and deficiency in 49% [34]. In our study, suboptimal levels (<30 ng/mL) were found in 75.7% of the control group, with a median of 22.9 ng/mL, which makes our findings consistent with those reported in the literature.

Studies indicate no significant differences in vitamin D deficiency prevalence between patients with CKD (stages 1–4) and the general population. Guesseous et al. reported relatively similar deficiency rates (69% in CKD stages 3–4 vs. 75% in the general population) with comparable mean 25(OH)D3 levels (23.1 vs. 23.5 ng/mL) [35]. Bosworth’s study showed that 25(OH)D3 levels do not decline with worsening eGFR due to balanced production and catabolism [36]. On the other hand, studies regarding dialysis patients show a significant decrease in 25(OH)D3 levels, reflected in the high prevalence of deficiency in French and Palestinian studies reporting rates of 89% and 87%, and severe deficiency (<10 ng/mL) in 42% and 22%, respectively [37,38], and in a study from Madrid that found suboptimal levels in 93% of dialysis patients [39]. According to some studies [40,41], CKD progression (stages 3 to 5D) is associated with declining vitamin D levels, likely due to reduced endogenous synthesis (limited sun exposure) and low-phosphate diets [42].

In our study, deficiency (<20 ng/mL) was found in 68.2% of HD patients and insufficiency (20–30 ng/mL) in 12.1%, resulting in suboptimal levels (<30 ng/mL) in over 80% of patients, which makes our findings consistent with those reported in the literature.

### 4.2. 24,25(OH)D3 and VMR

24,25(OH)D3 appears to be a very appropriate metabolite for assessing vitamin D metabolism disorders in CKD because it represents the result of a feedback mechanism. On the other hand, its relatively long circulating half-life, which is approximately 7 days, and its circulation at levels of 1–10 ng/mL, higher than any other vitamin D metabolites (except of 25(OH)D) and 100-fold higher than 1,25(OH)2D3, make it convenient for laboratory measurements [36,43].

We demonstrate a statistically significantly lower concentration of both 25(OH)D3 and 24,25 (OH)D3 as well as VMR in the study group, compared to the control group. However, the difference in concentrations between the general population and the HD patients is much greater in the case of the metabolite 24,25(OH)D3 than 25(OH)D3. A noteworthy observation is that in the HD group, for both 24,25(OH)2D3 and 3-epi-25(OH)D3, the correlation with 25(OH)D3 takes on a more logarithmic shape, whereas in the control group, the ratio of these metabolites appears to be more linear.

In our control group, the level of 24,25(OH)D3 was similar to that observed in the general population, as suggested by studies from the USA and South Korea, with a value of 2.09 ng/mL. The studies mentioned above reported levels of 2.7 ± 1.8 ng/mL and 1.9 ± 1.1 ng/mL, respectively [23,44].

Standardized reference ranges for these parameters (24,25(OH)D3 and VMR) are lacking, although some researchers have proposed a potential reference range for VMR, ranging from 4.4 to 14.3% [45], and from 0.4 to 8.9 nmol/L for 24,25 (OH)D3 [46]. The ratio of 24,25 (OH)D3 to 25 (OH)D3 (VMR) in the general population is approximately 10% (1:10). In individual studies, it ranges from 7.3% to 8.3%, and up to 12.3% [45,47] (Table 6).

Similarly, in our study, the VMR in the control group was comparable to those aforementioned and stated as 10.3%, whereas in the HD group, the median VMR level was significantly lower and equal to 0.97%. Moreover, a low VMR was reported in other studies regarding HD patients. In an American–Canadian study involving 91 HD patients, the median VMR was 1.28%, and the average levels of 24,25(OH)2D3 ranged from 0.15 to 0.55 ng/mL, which are also close to our results (median level: 0.1 ng/mL) [48]. Similar results were also presented by Weisman Y. et al. [49]. Renal CYP24A1 is mainly responsible for vitamin D catabolism and its activity depends on kidney function; therefore, decreased eGFR is associated with a reduction in circulating 24,25(OH)2D3 [36,50], whereas no significant correlation has been observed between 25(OH)D3 and eGFR [51]. Hence, the substantially lower VMR in the HD group compared to the general population primarily results from the difference in 24,25(OH)2D3 levels, as the differences in 25(OH)D3 between the populations are not as pronounced, as partly demonstrated in this study. This finding has also been corroborated in other studies, such as those by Gueossousa and Bosworth, as indicated above [35,36].

**Table 6 nutrients-17-00774-t006:** The reference values for vitamin D metabolites compared to the analyzed population [45,52,53].

Metabolite	Reference Value	Study Group [% *n* with Reference Range]	Control Group [% *n* with Reference Range]
25(OH)D3 [ng/mL]	>30 [ng/mL]	19.7% (*n* = 13/66)	24.4% (*n* = 50/205)
epi-25(OH)D3	no data existed	-	-
epi-25(OH)D3/25(OH)D3	no data existed	-	-
24,25(OH)2D3 [ng/mL]	>1.68 ng/mL (>4.2 nmol/L)	0% (*n* = 0)	62.0% (*n* = 127/205)
24,25(OH)2D3/25(OH)D3 (VMR)	4.4–14.3 [%]	1.8% (*n* = 1/56)	93.2% (*n* = 191/205)

### 4.3. 25(OH)D2

The clinical utility of measuring 25(OH)D2 is limited due to its functional overlap with 25(OH)D3 in the metabolic pathway. Armas et al. suggest that the shorter half-life of vitamin D2 reduces its effectiveness in treating 25(OH)D3 deficiency and preventing related diseases compared to vitamin D3 [3]. In most laboratories, 25(OH)D2 is not measured separately, and its concentration is reported together with 25(OH)D3. As a result, reference values for ergocalciferol are not typically available. Studies report average 25(OH)D2 concentrations of between 4.1 and 7.4 ng/mL, and low detectability (11–28%, respectively) [54,55]. In contrast to the above, our study found concentrations approximately ten times lower, likely due to a higher detection sensitivity (100% detection). Interestingly, 25(OH)D2 levels were higher in HD patients than the general population (0.61 ng/mL vs. 0.31 ng/mL), while other metabolites were lower. An interesting result was reported by Swanson et al. in that higher 25(OH)D2 levels correlated with lower 25(OH)D3 levels [56]. Similarly, Tang J. et al. observed higher 25(OH)D2 levels (G1–2: 1.6 ng/mL, G3a: 1.8 ng/mL, G3b+: 2.2 ng/mL) and decreasing 25(OH)D3 levels (16.4, 13.2, and 12.2 ng/mL, respectively) in advanced CKD [57]. This may reflect a greater stability or reduced clearance of vitamin D2 in comparison to D3 in patients with CKD. Additionally, it is noteworthy that patients with CKD, especially those with advanced disease, often follow a plant-based diet to limit phosphate intake, which may result in a proportionally higher consumption of plant-based products, further contributing to elevated 25(OH)D2 levels [58].

### 4.4. 3-epi-25(OH)D3

The clinical relevance of measuring 3-epi-25(OH)D3 remains uncertain due to the lack of standardized reference ranges for its levels or its ratio to 25(OH)D3 (Table 6). HPLC-MS (high-performance liquid chromatography–mass spectrometry), commonly used for 25(OH)D3 measurement, often fails to separate the C3 epimer, potentially inflating results and highlighting the need for improved methods and reference ranges [53].

Higher 3-epi-25(OH)D3 levels are observed in infants due to immature enzymatic systems and protection against hypercalcemia [19]. In the general population, the ratio of 3-epi-25(OH)D3 to 25(OH)D3 is ~5%, consistent with this study’s control group (mean 4.59%), and absolute concentrations average ~1.5 ng/mL (median 1.4 ng/mL), being consistent across various populations and closely aligning with our findings (median: 0.96 ng/mL) [18,19]. A Spanish study noted higher epimer levels (mean 4.5 ng/mL) in individuals with elevated 25(OH)D3 (>64 ng/mL), suggesting a proportional relationship [59], as seen in our study in both groups. Its protective value against excess vitamin D3 may also explain the positive correlation with 24,25(OH)D3, as demonstrated in this study (Table 5). Emerging research suggests that 3-epi-25(OH)D3 may have implications in CKD. Tang J. et al.’s study indicated a proportional increase in epi-25(OH)D3 concentrations with CKD progression (stages 2–4), although the values remained comparable to those of the general population [57]. Similarly, Arroyo E. et al., in a study on patients with advanced CKD, including kidney transplant recipients and those on dialysis, reported median concentrations ranging from 0.2 to 0.9 ng/mL [60], similar to this study’s HD group (0.4 ng/mL), and lower than the controls (0.96 ng/mL). Importantly, patients with CKD G5 on dialysis appear to being particularly susceptible to low levels of this metabolite. These findings, coupled with evidence linking 3-epi-25(OH)D3 to skeletal muscle strength, mass, and exercise capacity, suggest that monitoring this epimer may hold clinical significance in managing CKD-related complications [60].

### 4.5. 1,25(OH)D3

Although calcitriol reflects the metabolic activity of vitamin D, its routine measurement in CKD patients is not recommended by KDIGO due to its short half-life (4–6 h), high daily variability, and the impact of calcitriol or vitamin D analog supplementation on measurements. Moreover, the factors influencing the activity of the 25(OH)D3 1α-hydroxylase and 24(OH)D hydroxylase enzymes, along with the lack of standardized assay methods and evidence supporting its clinical utility, limit the practical application of this test [43,61]. The circulating level of 1,25(OH)2D3 is 1000 times lower than that of 25(OH)D3, making the accurate measurement of serum levels challenging. In this study, median serum 1,25(OH)2D3 concentrations were low (21.4 pg/mL; reference range: 25–45 pg/mL), consistent with previous studies reporting similar levels in dialysis patients (e.g., 20.1 pg/mL [62], and 22.4 pg/mL [63]). Patients in this study receiving alphacalcidol supplementation had significantly higher 1,25(OH)2D3 levels compared to those without supplementation, corroborating findings from other studies [64,65]. This increase highlights alphacalcidol’s therapeutic role in reducing excessive PTH secretion. Additionally, 1,25(OH)2D3 and 24,25(OH)D3 levels were positively correlated with serum 25(OH)D3 levels (Table 5), suggesting that maintaining adequate 25(OH)D3 levels contributes to higher levels of active vitamin D3 and its catabolite. Cholecalciferol supplementation may serve as an initial or partial alternative to alphacalcidol in dialysis patients (until the use of alphacalcidol becomes clinically necessary), improving vitamin D status and increasing 1,25(OH)2D3 levels, showing comparable or superior efficacy in bone disease prevention and SHPT management [62,65,66]. These findings highlight the potential of cholecalciferol for broader clinical benefits and enhanced safety in the long-term management of dialysis patients.

## 5. Conclusions

The measurement of 25(OH)D3 is the most accurate indicator of vitamin D status due to its long half-life. Still, it does not fully capture vitamin D’s metabolic activity, which is better reflected by 24,25(OH)2D3 and the VMR. Our study revealed significantly lower concentrations of 25(OH)D3 in the HD group, with a higher frequency of deficiency (69% vs. 39%). Despite this, both groups exhibited a similarly low frequency of optimal vitamin D3 levels (>30 ng/mL). Moreover, HD patients demonstrated lower levels of 24,25(OH)2D3, 3-epi-25(OH)D3, and VMR, compared to controls, indicating a functional deficiency in vitamin D3 that is more accurately captured than by 25(OH)D3 alone. Additionally, the significantly reduced 24,25(OH)2D3 levels observed due to the diminished activity of CYP24A1 in CKD reflect profoundly impaired vitamin D3 catabolism. These data suggest a reduced vitamin D reserve and a heightened demand for vitamin D in patients with CKD. Unlike other forms of vitamin D, 25(OH)D2 levels may increase in advanced CKD, reflecting different regulatory mechanisms requiring further investigation. Although calcitriol (1,25(OH)2D3) indicates vitamin D activity, its short half-life and variability limit routine use. Its positive correlation with 25(OH)D3 highlights the importance of maintaining adequate 25(OH)D3 levels. Cholecalciferol supplementation may also effectively manage SHPT as an alternative to alphacalcidol. However, a notable strength of our study is the inclusion of individuals who did not receive cholecalciferol supplementation. It is noteworthy that, to date, no study has specifically focused on 3-epi-25(OH)D3 among dialysis patients in comparison to other vitamin D metabolites. Additionally, our study introduces some new data to the limited number of studies on the end-stage kidney disease population regarding 25(OH)D2, 24,25(OH)D3, and the VMR.

Further research is necessary to better understand the clinical relevance and demonstrate the benefits of the above-mentioned metabolites as indicators of vitamin D status. At present, these markers are more of a research tool than a standard diagnostic measure.
The vitamin D3 reserves, assessed by 25(OH)D3 levels, were lower in the HD group than in the general population.Both functional deficiency and impaired vitamin D3 catabolism were present in the HD patients.The most sensitive parameter for assessing vitamin D3 deficiency was the VMR, which requires the measurement of 24,25(OH)D3.Alphacalcidol supplementation increases the concentration of 1,25(OH)2D3 without influencing 25(OH)D3.25(OH)D2 is the only studied vitamin D metabolite that reached higher concentrations in the HD group than in the general population.This study demonstrated a statistically significant positive correlation between 25(OH)D3 and 24,25(OH)2D3 as well as 3-epi-25(OH)D3 in both the hemodialysis and control groups, indicating a strong relationship between these metabolites in vitamin D metabolism regardless of renal function status.

### Limitations

Several limitations should be considered when interpreting this study’s results. The main limitations were the small number of HD patients, the variation in dialysis therapy duration, and the inclusion of some HD patients taking alphacalcidol (no statistically significant impact of alphacalcidol administration on metabolites other than calcitriol was observed). Another limitation of this study was the incomplete matching of demographic characteristics, including gender, between groups. Additionally, the groups differed significantly in terms of the number of participants, which may have influenced the statistical power of the analysis. However, the statistical analysis (Mann–Whitney test) indicated that the gender distribution did not significantly influence metabolite levels (*p* > 0.05), minimizing concerns regarding its potential confounding effect.

Dietary intake is another factor that was not controlled in this study, which may have influenced the levels of certain metabolites. Nutritional status and dietary restrictions in HD patients, particularly concerning calcium, phosphate, and vitamin D intake, can significantly affect biochemical parameters. However, due to the retrospective nature of the study, detailed dietary data were not available for analysis. Future research should consider the role of dietary factors in modulating these biochemical markers. Another limitation is that the study focused on biochemical parameters without evaluating clinical outcomes or conducting long-term observations.

Future research, especially on vitamin D supplementation and its long-term effects on bone and cardiovascular health, is necessary.

## Figures and Tables

**Figure 1 nutrients-17-00774-f001:**
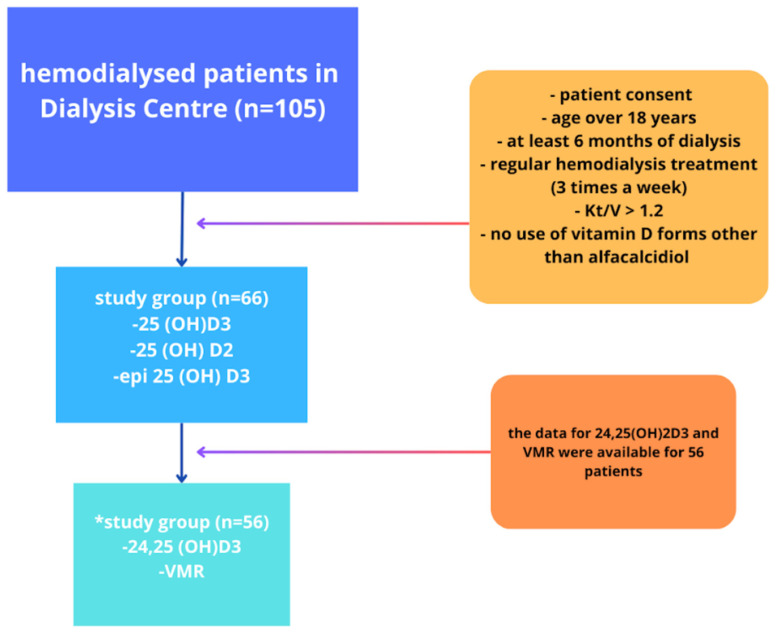
Criteria for participation in the study. * Results of the level of the metabolites 24,25(OH)D3 and 24,25(OH)2D3/25(OH)D3 (VMR were available for 56 patients).

**Figure 2 nutrients-17-00774-f002:**
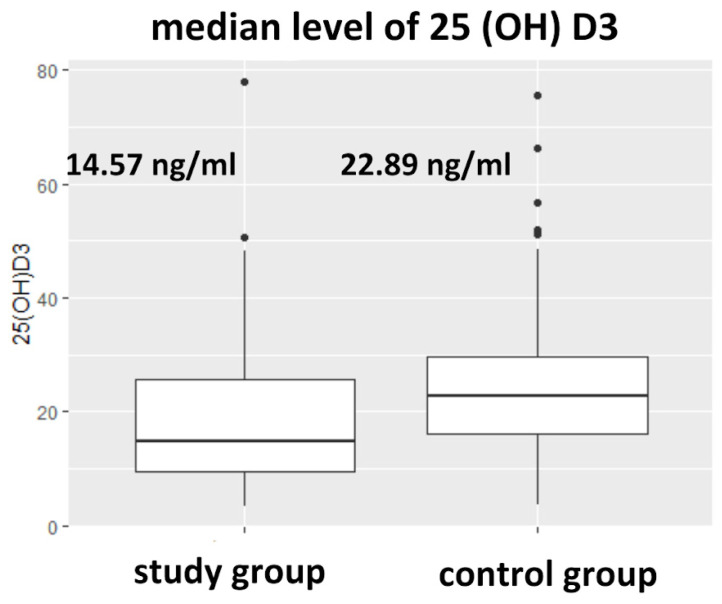
The difference in the level 25(OH)D3 between study (HD) and control population (*p* = 0.0001).

**Figure 3 nutrients-17-00774-f003:**
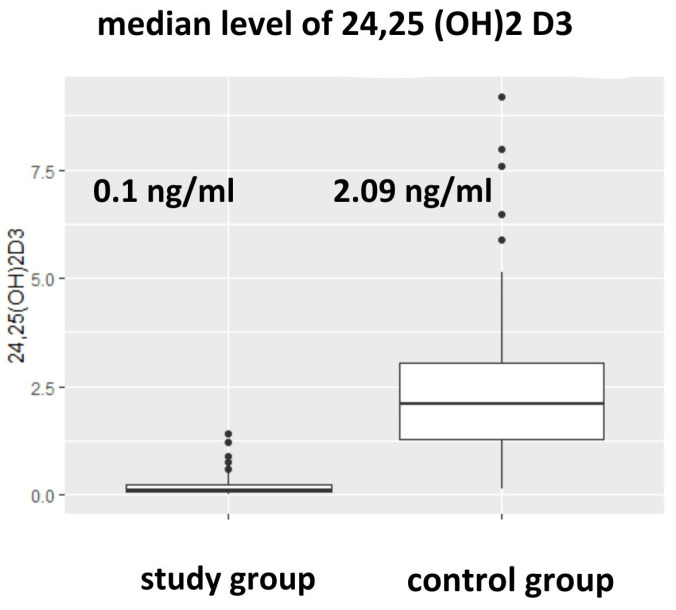
Difference in the level 24,25(OH)D3 between the study and control population (*p* = 0.0000).

**Figure 4 nutrients-17-00774-f004:**
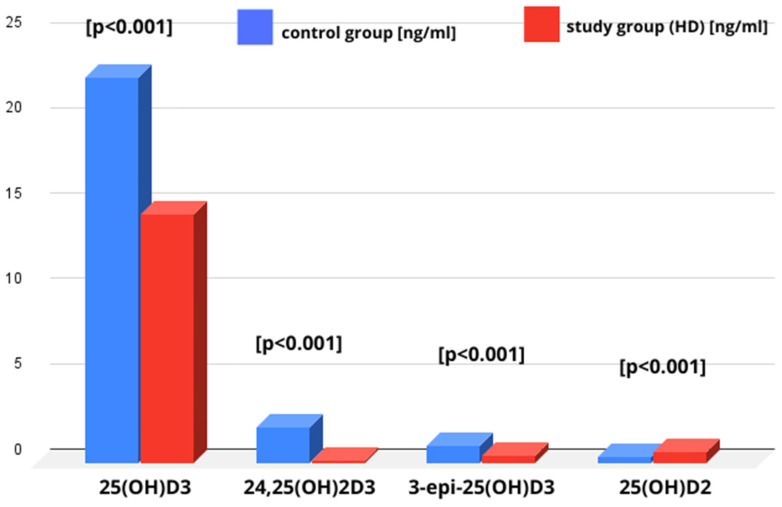
Comparison of vitamin D metabolite concentrations between the control group and the study group.

**Figure 5 nutrients-17-00774-f005:**
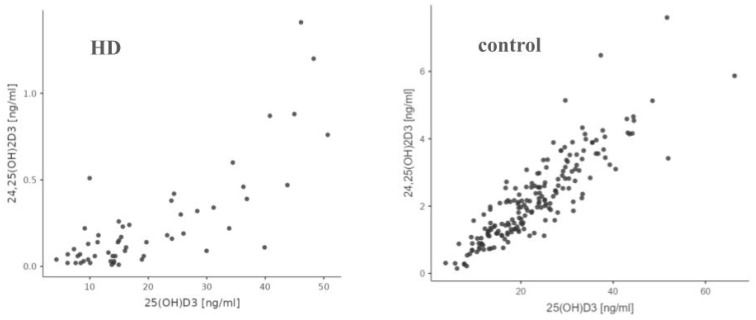
Comparison of correlation between 25(OH)D3 and 24,25(OH)2D3 in HD and control group.

**Figure 6 nutrients-17-00774-f006:**
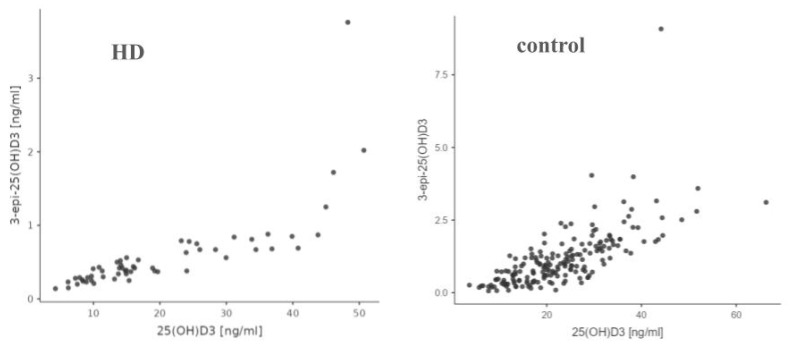
Comparison of correlation between 25(OH)D3 and 3-epi-25(OH)2D3 in HD and control group.

**Table 1 nutrients-17-00774-t001:** The characteristics of the study group and summary of the laboratory findings.

	HD Group *n* = 66	Reference Range
males *n*(%)	38 (57.6%)	
age [years] (average ± SD) (median)	61.3 ± 16.4 67	not applicable
Hgb [g/dL] (median) (Q1;Q3)	10.65 9.9; 11.6	12–15
therapy with erythropoetin *n*(%)	50 (75.8%)	not applicable
WBC [G/L] (median) (Q1;Q3) PLT [G/L] (median) (Q1;Q3)	6.45 5.5; 7.98 206 172; 262	4–10 150–410
residual diuresis (>500 mL/day) *n*(%)	24 (36.3%)	not applicable
Kt/V (median) (Q1;Q3)	1.63 1.46; 1.8	>1.2
CRP [mg/L] (median) (Q1;Q3)	4 2; 9.8	<5
Pi [mg/dL] (median) (Q1;Q3)	5.2 4.4; 6.3	2.5–4.5
Ca [mg/dL] (median) (Q1;Q3)	8.9 8.6; 9.5	8.5–10.2

**Table 2 nutrients-17-00774-t002:** Characteristics of the study and control group and summary of vitamin D metabolites. * the data for 24,25(OH)2D3 and VMR were available for 56 patients.

	All Participants *n* = 272	HD Group *n* = 66	Control Group *n* = 206	*p*-Value (Test M-W)
males (*n*%)	101 (37.1%)	38 (57.6%)	63 (30.6%)	0.001
age [years] (average ± SD median)	60.9 ± 14.4 65	61.3 ± 16.4 67	60.8 ± 13.7 65	0.807
25(OH)D3 [ng/mL] (median) (Q1;Q3)	20.80 13.55; 29.55	14.57 9.31; 25.27	22.89 16.17; 29.64	0.0001
25(OH)D2 [ng/mL] (median) (Q1;Q3)	0.39 0.24; 0.66	0.61 0.46; 0.93	0.31 0.21; 0.53	0.0000
epi-25(OH)D3 [ng/mL] (median) (Q1;Q3)	0.75 0.39; 1.42	0.40 0.29; 0.67	0.96 0.52; 1.6	0.0000
epi-25(OH)D3/25(OH)D3	3.72 [%]	2.77 [%]	4.59 [%]	0.0000
24,25(OH)2D3 [ng/mL] (median) (Q1;Q3)	1.50 0.73; 2.60	0.10 * 0.06; 0.31	2.09 1.30; 3.04	0.0000
24,25(OH)2D3/25(OH)D3 (VMR) (median) (Q1;Q3)	8.24% 1.28%; 9.82%	0.91% * 0.37%; 1.40%	9.21% 5.23%; 10.22%	0.0000

**Table 3 nutrients-17-00774-t003:** Vitamin D status categories and prevalence.

25(OH)D3	Control Group [*n*]/% *n* = 206	Study Group (HD) [*n*]/% *n* = 66
deficiency < 20 ng/mL	82/39.8%	45/68.2%
insufficiency 20–30 ng/mL	74/35.9%	8/12.1%
sufficiency 30–50 ng/mL	44/21.4%	12/18.2%
high supply 50–100 ng/mL	6/2.9%	1/1.5%
toxicity > 100 ng/mL	0	0

**Table 4 nutrients-17-00774-t004:** Comparison of the vitamin D metabolites in HD patients with and without alphacalcidol supplementation.

Metabolite	Alphacalcidol Supply; *n* = 25	Without Alphacalcidol; *n* = 41	HD Group; *n* = 66	*p*-Value
1,25(OH)2D3 [pg/mL]	30.4	16.2	21.4	*p* < 0.01
25(OH)D3 [ng/mL]	14.06	14.94	14.57	*p* > 0.05
24,25(OH)2D3 [ng/mL]	0.15	0.14	0.14	*p* > 0.05
VMR	0.87%	0.96%	0.91%	*p* > 0.05
3-epi-25(OH)D3	0.42	0.38	0.40	*p* > 0.05

**Table 5 nutrients-17-00774-t005:** Correlations between studied parameters according to Spearman’s coefficient.

Metabolite	HD Group (R, *p*-Value)	Control Group (R, *p*-Value)	Interpretation
24,25(OH)2D3 and 25(OH)D3	R = 0.714 *p* < 0.001	R = 0.885 *p* < 0.001	Statistically significant positive correlation in both groups.
25(OH)D2 and 25(OH)D3	R = −0.179 *p* = 0.150	R = −0.046 *p* = 0.513	No statistically significant correlation in either group.
3-epi-25(OH)D3 and 25(OH)D3	R = 0.915 *p* < 0.001	R = 0.776 *p* < 0.001	Statistically significant positive correlation in both groups.
24,25(OH)2D3 and 3-epi-25(OH)D3	R = 0.692 *p* < 0.001	R = 0.780 *p* < 0.001	Statistically significant positive correlation in both groups.
1,25(OH)2D3 and 25(OH)D3	R = 0.383 *p* < 0.003	lack of data for control group	Statistically significant positive correlation in HD group

## Data Availability

Data and materials are available after contact with the corresponding author.

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
