# Peer review of "Assessment of Vitamin D Metabolism Disorders in Hemodialysis Patients"

_nutrients, 2025, doi:10.3390/nu17050774_

Round 1
Reviewer 1 Report
Comments and Suggestions for Authors
In the manuscript entitled “Assessment of vitamin D metabolism disorders in the hemodialysis patients” the authorsanalyzed the levels of several vitamin D metabolites in patients with chronic kidney disease undergoing hemodialysis (HD) compared to a control group without kidney disease. HD patients had significant vitamin D3 deficiency, reduced 24,25(OH)D3 levels, and impaired vitamin D metabolism, evidenced by a lower vitamin D metabolism ratio (VMR). Alfacalcidol supplementation increased 1,25(OH)2D3 levels, but did not affect other metabolites. Additionally, HD patients had higher 25(OH)D2 levels compared to controls. VMR was found to be the most sensitive parameter for assessing vitamin D3 deficiency, highlighting the importance of measuring 24,25(OH)D3. The manuscript addresses a relevant topic regarding vitamin D metabolism in patients undergoing hemodialysis (HD). The data presented are significant and well structured, providing a detailed comparative analysis between the HD group and the control group. However, there are some areas where the work could be improved both in terms of clarity of presentation and experimental depth.
The manuscript is well written, but I add my suggestions below to improve it.
Minor points.
1. Although it is clear that the HD group and the control group were compared, it would be useful to provide more details on the inclusion and exclusion criteria of the participants.
The authors could specify whether the HD patients had a mean duration of dialysis and whether this could have influenced the levels of vitamin D metabolites.
2. The figures and tables are well referenced, but their interpretation could be improved with a more in-depth discussion of key data.
The use of means, medians, and interquartile ranges is appropriate, but it may be useful to also include scatterplots to better visualize the distribution of the data.
3. It would be interesting if the authors could measure levels of FGF23 (fibroblast growth factor 23), a key regulator of phosphate and vitamin D metabolism in patients with chronic kidney disease.
4. Additionally, it would be helpful if the authors could collect data on patients’ nutritional status, sun exposure, and diet, as these factors can influence vitamin D levels. If available, the authors should include information on liver function, as the liver plays a key role in the conversion of vitamin D.
Author Response
Comment 1.
Although it is clear that the HD group and the control group were compared, it would be
useful to provide more details on the inclusion and exclusion criteria of the participants.
The authors could specify whether the HD patients had a mean duration of dialysis
and whether this could have influenced the levels of vitamin D metabolites.
Response 1
Dear Reviewer
Thank you for your valuable comments. We agree that providing more detailed inclusion and
exclusion criteria, as well as information on the mean duration of dialysis in HD patients,
would enhance the clarity of our study. The limited scope of these data was due to our
concern about the already extensive length of the manuscript and our focus on essential
aspects of the research. However, in line with the reviewer’s suggestion, we will expand the
description of patient characteristics and supplement the inclusion and exclusion criteria with
the missing information that you find in the updated manuscript.
Comment 2.
The figures and tables are well referenced, but their interpretation could be improved
with a more in-depth discussion of key data.
The use of means, medians, and interquartile ranges is appropriate, but it may be
useful to also include scatterplots to better visualize the distribution of the data.
Response 2
Dear Reviewer,
This is a valid point. In response to the suggestion, we have generated visualizations
of the data distribution and incorporated them into the manuscript.
Comment 3
It would be interesting if the authors could measure levels of FGF23 (fibroblast growth
factor 23), a key regulator of phosphate and vitamin D metabolism in patients with chronic
kidney disease.
Response 3
Dear Reviewer,
We sincerely appreciate your insightful comment regarding the measurement of FGF23
levels. We fully acknowledge that FGF23 is a crucial marker in the regulation of phosphate
and vitamin D metabolism, particularly in patients with chronic kidney disease. Unfortunately,
at this stage, we do not have the possibility to include this parameter in our study.
Nevertheless, we share your opinion on its significance, and we believe that future research
incorporating FGF23 measurements would be highly valuable in further elucidating
disturbances in vitamin D metabolism. Such studies could contribute to enriching the global
literature in this broad and complex field.
Comment 4.
Additionally, it would be helpful if the authors could collect data on patients’ nutritional
status, sun exposure, and diet, as these factors can influence vitamin D levels. If available,
the authors should include information on liver function, as the liver plays a key role in the
conversion of vitamin D.
Response 4
Dear Reviewer,
Thank you for your valuable comments regarding the factors influencing vitamin D levels.
We would like to clarify that the patients in the study group, like all patients in our Dialysis
Center, were under the regular care of renal dietitian. The standard dietary
recommendations included phosphate intake of up to 1000 mg/day, protein intake of
approximately 1.0–1.3 g/kg body weight, and potassium intake of 2–2.5 g/day. The patients'
diet was based on the consumption of animal protein (meat, eggs, fish). This diet could be
adjusted based on the patient’s nutritional status and laboratory results, including urea,
potassium, albumin, and phosphate levels. Additionally, in our center, nutritional status is
assessed using the Subjective Global Assessment (SGA) every 6–12 months. None of the
patients met the criteria for severe malnutrition, which was an exclusion criterion for the
study.
We do not have detailed data on sun exposure; however, in the hemodialysis patient
population, UVB exposure is estimated to be approximately 50% lower compared to healthy
peers. This significant difference is primarily due to the fact that each dialysis session,
including transportation to the center, requires approximately 6 hours. Additionally, post-
dialysis discomfort often discourages patients from engaging in outdoor activities, meaning
that dialysis days are typically spent with minimal exposure to sunlight.
With regard to liver function, there is no single parameter that would allow for a direct
assessment of its efficiency, analogous to creatinine clearance in renal failure. However,
recognizing the liver’s critical role in vitamin D3 metabolism, we intentionally excluded
patients with evidence of liver insufficiency or chronic liver diseases from the study. The
updated exclusion criteria now explicitly reflect these considerations. It is also worth noting
that thrombocytopenia is a common laboratory abnormality in liver dysfunction. However, all
quartiles in our study population were within the normal range for platelet count, minimizing
concerns regarding potential liver-related hematological abnormalities.
Reviewer 2 Report
Comments and Suggestions for Authors
Comments to the authors
This is a cross-sectional study aiming to compare the vitamin d status and metabolism between hemodialysis patients and control without CKD. I have the following comments to the authors:
1. The control group is not similar to the dialysis group. For example, there is a significant difference in the gender. A case-control design and a matching procedure for some basic clinical characteristics is important.
2. Table 1: several important clinical and laboratory characteristics of the patients enrolled in the study are not presented (co-morbidities, basic labs, other treatments)
3. The interventional part of the study is not clear. How ere these patients selected? Id you include a control group treated with placebo? There is no randomization also.
4. Please justify the decision to use alfacacidol (active vitamin d) and not nutritional vitamin d supplementation (cholecalciferol or ergocalciferol).
5. The limitations of this study need to be adequately addressed in the discussion.
6. The results are rather confirmatory – please explain in detail the novelty and the rationale of your study.
Author Response
Comment 1.
The control group is not similar to the dialysis group. For example, there is a
significant difference in the gender. A case-control design and a matching procedure for
some basic clinical characteristics is important.
Response 1
Dear Reviewer,
Indeed, the groups differ significantly in terms of gender distribution. To assess whether
this impacts the presented results, I applied statistical methods (Mann-Whitney test) for
each metabolite in both the control and experimental groups, using gender as the
grouping variable. A statistically significant difference (p<0.05) was found only for
25(OH)D2 in the experimental group. Therefore, I can include this information in the
manuscript.
* In response to the need for partial reanalysis to demonstrate that gender does not have
an impact on metabolite levels, one mistake regarding the correlation between 3-epi(OH)D3
and 25(OH)D3 in the HD group was identified and corrected. The correlation is now
positive and statistically significant, as observed in the control group.
Comment 2
Table 1: several important clinical and laboratory characteristics of the patients
enrolled in the study are not presented (co-morbidities, basic labs, other treatments)
Response 2
Dear Reviewer,
Thank you for bringing this to our attention. As authors, we also had some concerns
regarding this aspect (basic labs). Ultimately, we decided to present a concise
manuscript to ensure clarity and readability for the reader. However, if the reviewer
considers it necessary, we will make the required modifications.
We added to the Methods section information about primary disaeses cause CKD in HD
group.
Nevertheless, chronic conditions that may significantly impact vitamin D metabolism—such
as liver failure, liver diseases, disease affecting calcium- phosphate imbalance, autoimmune
diseases requiring immunosuppressive therapy, and advanced-stage cancer—were
excluded from the study
Comment 3.
The interventional part of the study is not clear. How were these patients selected? Id
you include a control group treated with placebo? There is no randomization also.
Response 3
Dear Reviewer,
Thank you for your insightful comments on our manuscript. We would like to address your
question regarding the intervention and control group.
We wish to clarify that this study was observational (real life study) and not an interventional
study. Its aim was to investigate whether there is a relationship between the use of
alfacalcidol (for at least 3 months) and the levels of individual vitamin D metabolites. We
wanted to identify the potential influence of alfacalcidol on the levels of 24,25(OH)2D3, epi-
25(OH)D, and 25(OH)D2, in order to exclude this factor as a potential confounder when
comparing these metabolites between the studied populations.
Comment 4.
Please justify the decision to use alfacacidol (active vitamin d) and not nutritional
vitamin d supplementation (cholecalciferol or ergocalciferol).
Response 4
Dear Reviewer,
At the time of conducting the study, the previous KDIGO 2017 guidelines were in effect
(the current ones are from 2024), according to which supplementation with the active
form of vitamin D3 was recommended for the treatment of SHPT, especially in dialysis
patients. In our dialysis center at that time, cholecalciferol was practically not used
among patients, only active forms of vitamin D3 remained, including alfacalcidol.
At the time the study was conducted, the KDIGO 2017 guidelines were in effect (the
current ones date from 2024), which recommended supplementation with the active form
of vitamin D3 for the treatment of secondary hyperparathyroidism (SHPT), particularly in
dialysis patients. Our approach was therefore fully aligned with these prevailing
recommendations. Additionally, in our dialysis center at that time, cholecalciferol was
rarely used (these patients were excluded from our study), and patients predominantly
received active vitamin D3 formulations, including alfacalcidol. This reflects a real-life HD
patient population worldwide, in which individuals with SHPT requiring treatment with
active vitamin D3 constitute a significant proportion of dialysis patients. Excluding them
from the study could have substantially reduced the study group and its clinical
relevance. On the other hand, as explained in the original manuscript, we demonstrated
statistically that alfacalcidol supplementation had no significant impact on the vitamin D3
metabolites in the HD group, as staid: “However, there was no statistically significant
difference in the median concentrations of the metabolites: 25(OH)D3, 24,25(OH)2D3,
epi3-25(OH)D3, and the 24,25(OH)2D3/25(OH)D3 ratio between these patients [table 4].
Therefore, we compared metabolites in the HD group to that of the general population,
despite some patients using alfacalcidol. This approach is justified as the
supplementation did not influence the metabolite concentrations, allowing for a unified
analysis across the dialysis cohort. “
Comment 5. The limitations of this study need to be adequately addressed in the discussion.
Response 5
Dear Reviewer,
We have addressed the reviewer’s suggestion and expanded the Limitations section to include
additional details. Rather than incorporating these points into the broader Discussion section,
we believe it is more transparent and reader-friendly to present all study limitations in one
dedicated section. This approach ensures clarity and prevents key considerations from being
fragmented across different parts of the text.
Comment 6. The results are rather confirmatory – please explain in detail the novelty and the
rationale of your study.
Response 6
Dear Reviewer,
Thank you for your insightful comments on our manuscript. We acknowledge the perceived
redundancy in certain sections of our work. However, as both authors and practicing
clinicians, we understand the critical importance of reiteration when it comes to data utility
and safety. By reinforcing key information ("one swallow doesn't make a summer"), we aim
to ensure that our findings are easily understood and can be confidently applied in clinical
practice, ultimately contributing to improved patient care. We have carefully considered your
feedback and believe that the current level of repetition is necessary to achieve this goal.
Nevertheless, we would like to emphasize certain novel aspects that distinguish our study.
1.Use of VMR as the most sensitive biomarker for vitamin D3 deficiency
● Previous studies have focused mainly on 25(OH)D3 and 1,25(OH)2D3 as markers
for vitamin D status in hemodialysis (HD) patients but VMR as a primary marker for
deficiency is less frequently highlighted and our study emphasizes VMR
(24,25(OH)D3/25(OH)D3 ratio) as the best indicator.
● Our work contributes a new methodological insight by demonstrating the higher
sensitivity of VMR, a metric that is not yet standard in clinical practice.
2. Different metabolism of 25(OH)D2 in HD patients in comparison to general
population.
● Our findings that 25(OH)D2 levels are higher in HD patients than in the general
population is also novel, for example recent works did not involve comparison with
the general population.
● No clear studies were found comparing 25(OH)D2 levels between hemodialysis
patients and the general population; most studies described all CKD patients, not
only HD patients.
● Most studies focus on overall vitamin D metabolism disturbances in CKD patients
rather than direct comparisons of 25(OH)D2 levels.
3. Assessment of epimeric vitamin D in HD population.
At the time we began writing this manuscript, there was a scarcity of research regarding the
concentration of this metabolite in the dialysis patient population. While a study by Campos
et al. (2024) has recently been published, our research independently confirms that
hemodialysis patients exhibit reduced levels of 3-epi-25(OH)D3, an observation that has not
been extensively explored previously. Our study distinguishes itself by specifically examining
the association of this metabolite within the dialysis population, a crucial aspect not
addressed in their work. Studies often discuss epimers separately, but their consistent
correlation with 25(OH)D3 across different populations is less frequently examined.
4. Strong correlations between 25(OH)D3 and both 24,25(OH)D3 and epimeric
25(OH)D3:
While some studies report correlations between these metabolites, our finding that
these correlations are statistically significant in both control and HD groups is
potentially novel.
5. An important advantage of our work is that it provides a comprehensive
comparison with the general population, whereas previous studies have mainly
concentrated on HD patients or other select groups.
We appreciate your understanding and look forward to your positive consideration of our
manuscript. We hope this explanation clarifies your concerns. Thank you again for your
valuable comments, which will help us improve our manuscript.
Kinds regards
Round 2
Reviewer 2 Report
Comments and Suggestions for Authors
No additional comments.